# *Cryptococcus neoformans* Genotypic Diversity and Disease Outcome among HIV Patients in Africa

**DOI:** 10.3390/jof8070734

**Published:** 2022-07-15

**Authors:** Kennedy Kassaza, Fredrickson Wasswa, Kirsten Nielsen, Joel Bazira

**Affiliations:** 1Department of Microbiology and Parasitology, Mbarara University of Science and Technology, Mbarara P.O. Box 1410, Uganda; kkassaza@must.ac.ug (K.K.); fredricksonwasswa@gmail.com (F.W.); 2Department of Microbiology and Immunology, University of Minnesota, Minneapolis, MN 55455, USA

**Keywords:** *Cryptococcus neoformans*, genetic diversity, fungal disease, cryptococcosis, cryptococcal meningitis, HIV-associated, genotypic, advanced HIV AIDS

## Abstract

Cryptococcal meningoencephalitis, a disease with poor patient outcomes, remains the most prevalent invasive fungal infection worldwide, accounting for approximately 180,000 deaths each year. In several areas of sub-Saharan Africa with the highest HIV prevalence, cryptococcal meningitis is the leading cause of community-acquired meningitis, with a high mortality among HIV-infected individuals. Recent studies show that patient disease outcomes are impacted by the genetics of the infecting isolate. Yet, there is still limited knowledge of how these genotypic variations contribute to clinical disease outcome. Further, it is unclear how the genetic heterogeneity of *C. neoformans* and the extensive phenotypic variation observed between and within isolates affects infection and disease. In this review, we discuss current knowledge of how various genotypes impact disease progression and patient outcome in HIV-positive populations in sub-Saharan African, a setting with a high burden of cryptococcosis.

## 1. Introduction

*Cryptococcus neoformans* causes cryptococcal meningitis (CM) and is a major cause of mortality throughout the developing world, especially among individuals living with advanced HIV/AIDS [1]. Cryptococcosis is a common AIDS-defining illness and a leading cause of mortality among adults with HIV [1]. Despite the advent of antiretroviral therapy, which drastically reduced the number of HIV cases in the developed world, CM remains a major problem in resource-limited regions [2].

Due to the burden of HIV in Africa, CM is the most common cause of adult meningitis in Sub-Saharan Africa, with 70% of all cases of CM globally occurring in sub-Saharan Africa [1,2]. Survival after cryptococcosis in sub-Saharan Africa is often ≤40% [3,4]. A number of clinical adverse prognostic markers in HIV-associated CM have been identified, including high fungal burden at CM diagnosis, poor rate of cryptococcal clearance from patient cerebrospinal fluid (CSF) during antifungal treatment, and altered mental status at presentation [5]. Patient-to-patient differences in clinical phenotype likely reflect a complex interplay between host factors (level of immunosuppression, immune response phenotype [6]), pathogen virulence [7], and health system factors such as delays in diagnosis and treatment [8,9]. Furthermore, long-term natural selection of *C. neoformans* within individuals by human antimicrobial defenses is proposed to occur [10,11], with the resultant likelihood that virulence factors will demonstrate natural variation within and amongst lineages [5].

Previous studies in other pathogenic microorganisms and fungi have shown evidence of genetic lineages associated with strain phenotype and clinical outcome [12,13]. Both multi-locus sequence typing (MLST) [5,14,15] and genome-wide association studies (GWAS) [7,16] with isolates from Uganda and Malawi provide the first evidence for differences in *Cryptococcus neoformans* virulence across clades in both humans and mouse models of cryptococcosis. In other studies analyzing South African clinical isolates, patient survival was associated with clinical isolate macrophage phagocytosis and mitochondrial fragmentation [17]. Finally, both small and large-scale variation, including aneuploidy, is associated with alternate growth phenotypes that may impact the course of infection [18].

## 2. Molecular Classification of *Cryptococcus* spp.

Cryptococcus is a genus of basidiomycetous fungi with more than 30 species found in the environment. Within this genus, a number of species are known to cause human disease, with the species most commonly associated with human disease being *Cryptococcus neoformans*, *Cryptococcus deneoformans* and the five species that compose the *Cryptococcus gattii* species complex (Figure 1) [19]. Recently proposed taxonomy based on molecular and genetic studies divided the major human pathogenic Cryptococcus species from the original single *Cryptococcus neoformans* designation into these seven more clearly defined species. At present, these seven major human pathogenic Cryptococcus species are sub-divided into two species complexes: the *Cryptococcus neoformans* species complex that contains *C. neoformans* (serotype A; genotypes VNI, VNII, VNB) and *C. deneoformans* (serotype D; genotype VNIV) and the *Cryptococcus gattii* species complex that contains an additional five species—*C. gattii*, *C. bacillisporus*, *C. deuterogattii*, *C. tetragattii*, and *C. decagattii* (serotypes B and C; genotypes VGI-IV) (Table 1) [20,21]. The molecular taxonomy of the Cryptococcus genus is a vibrant area of research that is enhancing our understanding of specific strain characteristics, including fitness, predilection for certain environmental niches, and association with human disease outcomes. Several molecular methods are used for taxonomic analysis of the *C*. *neoformans* and *C*. *gattii* species complexes, including restriction fragment length polymorphism (RFLP), microsatellite fingerprinting, multi-locus sequence typing (MLST), and whole-genome sequencing (WGS) [22,23,24,25]. Genome sequencing has identified only minor discordance between phylogenies produced using the previously defined VN/VG clade designations or MLST-defined sequence types (STs); thus, the field largely uses VN/VG or sequence type as a standardized system to classify genotypes within the human pathogenic *Cryptococcus* spp.

Most clinical and environmental isolates within the human pathogenic *Cryptococcus* spp. are haploid, although diploid and aneuploid isolates are observed [26,27,28]. Genomes range in size from 16 to 19 Mb and contain variable numbers of chromosomes. The *C. neoformans*, *C. deneoformans*, and *C. gattii* species complex reference strains all have 14 chromosomes along with the mitochondrial genome [29]. The human pathogenic *Cryptococcus* spp. have a bipolar mating system encoded by the MAT locus, with strains designated as either MATa or MATα [30,31]. The two mating types are morphologically similar in appearance and thus must be distinguished using molecular methods (e.g., PCR, sequence analysis, etc.) or by using a mating assay [28,32]. Interestingly, MATα isolates predominate and this mating type also appears to undergo monokaryotic mating more readily than MATa isolates, possibly explaining the reason for their increased prevalence [28].

## 3. African *C. neoformans* Strains Have Unique Characteristics

There are a number of possibilities for the high mortality rates observed with African cryptococcosis. The rates of cryptococcosis in Africa are consistent with HIV prevalence in this region of the world [1,33], showing a positive association between HIV and the prevalence of *C. neoformans* co-infection. However, along with high incidence, mortality in sub-Saharan Africa is also higher than many other regions of the world. Currently, the reason for this higher mortality is unclear. Clinical trials performed in Uganda that utilize the same standard-of-care as in the United States show lower mortality rates compared to general care, suggesting that clinical practices in sub-Saharan Africa may impact patient mortality [34,35,36]. Similarly, flucytosine, which is recommended to be used in conjunction with amphotericin B, is not available currently in much of sub-Saharan Africa [37,38]. The regimen of amphotericin B and fluconazole that is typically used in Africa was recently shown to be inferior to the amphotericin and flucytosine regiment [39,40]. Interestingly, this study found that the completely oral regimen of flucytosine and fluconazole was as effective as amphotericin B and flucytosine in African patients [40]. These data were surprising and suggest that optimal drug treatment strategies for African patients should be explored further [38].

Whether there is a genetic basis for the high cryptococcosis mortality rates in Africa is less well defined. Human genetic factors that influence cryptococcosis have been identified [19,41], but whether these genetic factors could account for the increased mortality observed in African patients has not been extensively investigated. Instead, early reports suggest that differences in *C. neoformans* genotypes could contribute to the high mortality observed in Africa [42]. A study performed by Litvintseva et al. on isolates from 200 HIV-seropositive patients in Botswana identified novel genotypes that differed from isolates found globally and thus were referred to as the VNB lineage. Analyses of mating types in this population indicated that 12% of these strains possess the MATa mating type, higher than among non-African isolates [43]. While it was originally proposed that this VNB lineage was geographically confined to sub-Saharan Africa, later studies identified VNB isolates in other regions of the world [43,44]. Interestingly, while the global VNI and VNII lineages appear to be highly clonal, the VNB lineage is not and led to speculation that sub-Saharan Africa could be the origin of *C. neoformans* [5,45].

Importantly, other common sequence types found in Africa can be globally distributed. For example, the most common sequence type in Uganda, ST93, is also frequently observed in South America, where it is also associated with high mortality rates [46]. In a study in South African pediatric patients, the most prevalent sequence type (ST8) was also global. In South Africa, ST8 was associated with male patients, the isolates exhibited high genetic diversity, and a high percentage of the strains were diploid [39,47]. In a study carried out in Yaoundé, Cameroon, 24% of HIV-infected patients with cryptococcal meningitis co-infections had infection with multiple genotypes [48]. At least two isolates with different antifungal susceptibilities were identified within a single patient sample, despite lack of antifungal treatment prior to sample collection [49]. These studies highlight the increased rates of genetic variability in Africa, even within globally distributed sequence types. In vivo microevolution has been reported and the frequency of mixed infections in African patients, whether due to microevolution or co-infection, may be higher than reported globally [50,51].

## 4. Disease Manifestations and Epidemiology

Successful disease initiation and progression likely rely on numerous genotypic and phenotypic factors of both the host and the fungus. More simply, a host must be susceptible and exposed to a Cryptococcus isolate that is sufficiently pathogenic before disease can occur. Most human exposure begins with inhalation of aerosolized cells, most likely spores, from the environment [52] into the lungs where the yeast cells are either cleared by the immune system or establish a latent pulmonary infection [53,54,55,56,57,58]. The timing of this initial environmental exposure to Cryptococcus may vary by geographic region and may depend on other socio-cultural factors, but by adulthood approximately 70% of people have developed antibodies to the *C. neoformans* species complex [59,60,61].

Infections with Cryptococcus are predominantly classified based on the site of infection. Clinically, cryptococcosis typically presents as cryptococcal meningitis (CM), although pulmonary or disseminated cryptococcosis is also frequently observed. Recently, classification based upon Cryptococcus species complex has also become more prevalent due to an outbreak of *C. deuterogattii* in North America and Europe, but a better understanding of the role that Cryptococcus species play in disease epidemiology is needed [62]. While latent pulmonary cryptococcosis is the most common infection due to Cryptococcus, it is predominantly asymptomatic [63]. Cryptococcus is also one of the yeasts most frequently described as members of the pulmonary mycobiome, as demonstrated in a study by Rubio-Portillo et al. [64]. Thus, initial pulmonary infection is acquired almost exclusively from the environment via inhalation of infectious, aerosolized basidiospores or desiccated yeast cells [52,65,66]. Extra-pulmonary infections are thought to be secondary to the primary pulmonary infection, even in cases where the latter is not readily evident. Disseminated cryptococcosis, particularly to the central nervous system (CNS), where it can produce meningoencephalitis and cryptococcomas during CM, is often observed in severely immunocompromised individuals. Cryptococcosis is classically seen in patients with advanced HIV and/or in individuals with CD4+ T-cell counts below 100 [67].

Susceptible hosts may experience an asymptomatic latent pulmonary infection that becomes acute pulmonary cryptococcosis (PC) during an immunosuppressive event and/or disseminate throughout the body to the CNS to ultimately cause cryptococcal meningitis [67,68] Alternatively, in a host that is susceptible upon exposure to the yeast, acute infection may manifest and disseminate without a latent stage. Current theories propose that many of the traits that promote Cryptococcus survival within its environmental niche also act as virulence factors in humans and contribute to fungal survival, disease initiation, immune evasion and dissemination of the infection from the lungs to the predominant site of disease in the brain [69].

In general, Cryptococcus preferentially localizes to the lungs and brain during infection; however, most organs have been reported as secondary sites of infection (e.g., skin, prostate) due to dissemination [70,71,72,73]. The epidemiology of human pathogenic *Cryptococcus* spp. has been studied since the 1980s. Disease due to the *C. neoformans* species complex is predominantly observed in immunocompromised individuals, but is also observed in some individuals that have no known immune deficiencies [74]. The *C. gattii* species complex, conversely, was historically regarded as a pathogen of apparently immunocompetent patients. However, pre-existing conditions and immunocompromised states, including subclinical immune defects, are also frequently observed in patients with *C. gattii* infections [75,76,77]. Thus, it is unclear whether a better understanding of the subpopulations within each species will explain the apparent patient variability, or whether the species differences in clinical presentation are primarily determined by variable host predilections [78,79,80].

As described in a recent review by Altamirano et al. [46], the various *Cryptococcus* spp. also display different disease epidemiology and clinical manifestations. *C. neoformans* is the most common species to cause infections globally, accounting for 95% of infections overall, and 99% of infections in individuals with advanced HIV [81]. While the majority of *C. neoformans* infections occur in patients with an immunocompromising condition, infections by the *C. gattii* species complex predominantly occur in immunocompetent individuals [46]. CM is a common disease manifestation in patients with *C. neoformans* species complex infections, with over 80% of patients displaying meningitis symptoms [82]. In contrast, CM is less common in *C. gattii* species complex infections, with patients typically presenting with pneumonia [79,83]. While rare globally, *C. deneoformans* infections are more frequently observed in Europe, often in the context of hybrids with *C. neoformans*, and in approximately 14% of infections where skin lesions are observed [84].

While less frequent, a few other species of *Cryptococcus* have been documented to cause disease in severely immunocompromised individuals. *Cryptococcus laurentii* is associated with fungemia, lung abscesses, and meningitis [85]. *Cryptococcus albidus* is another very rare species associated with ocular infections and meningitis [86,87]. *Cryptococcus uniguttulatus* is associated with ventriculitis and was first isolated from a human nail [88]. Given that the vast majority of human infections are caused by *C. neoformans*, we will predominantly focus our discussion on this species.

## 5. Host–Pathogen Interactions during Cryptococcus Infection

Many fungal species kill mammalian tissue culture cells upon in vitro co-culture [89,90]. Surprisingly, Cryptococcus exhibits minimal toxicity to mammalian cells in culture, leading to the suggestion that growth within mammalian cells may be beneficial for fungal cell survival or dissemination [91,92]. IFNγ-producing CD4+ T-cells are required for the activation of myeloid cells, especially macrophages, to enable fungal killing and clearance. However, macrophages may also act as a reservoir of the fungal yeast cells, shielding them from host immune detection and thus promoting latent infection or persistent chronic inflammation [93]. In a previous study, macrophage cell lines with phagocytosed *C. neoformans* were capable of growth and cell division, with the fungal cells able to transmit to the daughter macrophage [94]. These in vitro studies lead to the hypothesis that some *C. neoformans* isolates do not produce high levels of cytotoxic factors, promoting their survival within the phagocyte.

The outcome of the *C. neoformans*–macrophage interaction is a critical determinant for the fate of the pathogen and host during infection. The ability of *C. neoformans* to replicate inside macrophages correlates with infection susceptibility in animal models [95]. Similarly, the capacity of *C. neoformans* isolates to replicate in macrophages is correlated with worse human clinical outcomes [17,96]. These data suggest that factors and interventions that modulate macrophage function, especially in patients where adaptive immunity from T-cell function is impaired, could reduce disease, whereas the capacity of the pathogen to efficiently replicate intracellularly could be associated with progression of infection. Reduced macrophage activation impairs the antifungal capacity of these cells, which in turn facilitates intracellular Cryptococcus growth [97,98]. Finally, damage to mitochondria, reduced phagosome maturation, and induction of programmed cell death pathways in host cells during Cryptococcus infection directly aid in fungal cell survival in vivo and in vitro [99].

Consistent with these observations, Cryptococcus infections are not associated with large amounts of tissue necrosis, in contrast to infections caused by other fungal pathogens such as *Aspergillus* spp. or Mucorales. Instead, cryptococcosis infections tend to have features consistent with a chronic infection, with host death frequently resulting from physical compression of tissue, such as meningoencephalitis, or the presence of fungal masses, called cryptococcomas, in the brain that are associated with minimal to no inflammation [97]. The cryptococcomas have a distinctive appearance in magnetic resonance imaging that are sometimes referred to as “soap bubbles” and are composed of gelatinous pseudo cysts containing Crytococcus cells with large amounts of capsule polysaccharide. The cryptococcomas displace or destroy brain tissue to create the space for the fungal mass. Combined with the observation that *C. neoformans* replicates inside host cells, the cryptococcomas may be the result of progressive lysis of host cells during the host–pathogen interaction [97]. Alternatively, the cryptococcomas may compress surrounding brain tissue [100]. Defects in the resorption of the cerebrospinal fluid (CSF), thought to be due to its increased viscosity due to the presence of the capsule polysaccharide being released into CSF, are also known to cause overwhelming brain edema [101].

These observations lead to the conclusion that *C. neoformans* infections are associated with minimal host damage, but several reports indicate that *C. neoformans* is able to cause direct damage to host cells and tissues, with the damage attributed to both the fungus and the host immune response. For example, Immune Reconstitution Inflammatory Syndrome (IRIS), an exaggerated inflammatory response causing a subset of persons with recent CM to deteriorate with improving immune function, often occurs in the absence of culturable fungus and is due to inappropriate immune system activation in response to residual Cryptococcus antigen [102,103,104,105,106,107].

The blood–brain barrier (BBB) is a highly selective semipermeable border of endothelial cells that prevents solutes in the circulating blood from non-selectively crossing into the extracellular fluid of the central nervous system [108]. For Cryptococcus to invade the central nervous system, it must cross the BBB. Studies have provided evidence that *C. neoformans* crosses the BBB using at least three mechanisms—active transcytosis, passive transcytosis, and within host cells using a Trojan Horse-like mechanism. During active transcytosis, the Cryptococcus cells induce uptake by the BBB endothelial cells, crossing this cell layer without damaging the BBB integrity [109,110]. Alternatively, in passive transcytosis, the fungal cells are trapped in the brain capillaries because of their size, resulting in a lesion that ruptures the capillary and disrupts the BBB integrity [110,111,112]. In Trojan Horse BBB penetration, the Cryptococcus cells are first phagocytosed by monocytes or macrophages, and then are thought to transit across the BBB within the phagocyte [113,114,115]. It is currently unclear whether a fourth mechanism for BBB penetration exists that requires an interaction with host phagocytes, but this interaction does not occur at the BBB [115,116].

Thus, disease is likely a complex balancing act between fungal virulence potential and host susceptibility [97]. Several mechanisms of host–pathogen interaction come into play to cause maximum damage during infection. These host–pathogen interactions have been previously reported to occur at molecular, cellular, tissue and organism levels. Damage at the molecular level in *C. neoformans* infections has been shown to be the result of secretion of various enzymes, such as proteases, nuclease, urease, and phospholipase, that result in degradation of host molecules, such as antibodies, and/or modification of cell membranes [97]. At the cellular level, host damage involves modification of host cellular compartments, fungal cell shape, organelles and accumulation of fungal materials in the cell leading to cellular damage. At the tissue level, disruption of host intracellular organization and accumulation of fungal cells leading to the creation of fungal masses has been reported. Finally, at the organismal level, damage is due to fungal growth and dissemination and the host immune response that often leads to intracranial hypertension ending in death [97]. A damage–response framework for Cryptococcus pathogenesis has also been proposed where disease occurs at one extreme due to lack of appropriate immune response and at the other extreme when aberrant or excessive host immune responses directly cause host damage and exacerbate disease [117].

## 6. *C. neoformans* Virulence Factors/Mechanisms

Extensively characterized in vitro, the classic Cryptococcus virulence factors include the polysaccharide capsule, melanin formation, growth at host body temperature, and secretion of enzymes such as phospholipase, laccase, and urease [10,53]. Many of these virulence factors are involved in the inhibition of phagocytosis or survival within phagocytes [118,119]. Thought to be a result of its role in confounding host defenses, the polysaccharide capsule has been shown to be a major and essential virulence factor. At least 35 Cryptococcus genes are needed for capsule synthesis [120]. Capsule formation involves carbohydrate metabolism that turns sugar into the polysaccharide backbone of the capsule [120]. Once the basic polysaccharide backbone is generated within the cell, the capsule polysaccharide is excreted and attached to the cell wall, requiring at least 40 additional gene products [120]. Owing to the critical role of capsule in virulence, and the fact that many genes are involved in capsule synthesis and secretion/attachment, these capsule genes are under active investigation.

Because the polysaccharide capsule is crucial for virulence and sugars are an important precursor for capsule formation, sugar intake genes are also associated with virulence. Recently, researchers have uncovered an important role for a specific sugar alcohol, inositol, in *C. neoformans* virulence. Mutant *C. neoformans* strains deficient in two inositol transporter genes (*ITR1* and *ITR3*) exhibited a reduced ability to cross the blood–brain barrier both in vitro and during in vivo animal model studies [111]. Further, a microarray study of the response of wild-type Cryptococcus cells to inositol treatment revealed overexpression of genes for breaking down inositol [111]. Studies with *itr1aΔ itr3cΔ* mutants in a mouse model of cryptococcosis showed an increased immune response compared to infection with the wild-type strain [121]. These inositol mutants have decreased capsule production, indicating that inositol is a critical building block for proper capsule production [121].

In addition to polysaccharide, other cellular components are embedded in the capsule. Two components of the capsule shown to be important for virulence are mannoproteins and hyaluronic acid. T-cells obtained from mice immunized with *C. neoformans* mannoprotein proliferate in vitro when stimulated with mannoprotein in the presence of antigen-presenting cells (APC). When separated by SDS-PAGE, the fraction with an apparent molecular mass of >60 kDa contains the majority of the stimulatory activity [122]. Cryptococcal mannoproteins account for a large percentage of the secreted and cell-associated material from *C. neoformans* and are thus likely to be encountered frequently during the course of a cryptococcal infection [115]. In preliminary experiments by Levitz et al., C57BL/6 mice immunized with cryptococcal mannoproteins are partially protected from a subsequent intravenous challenge with live *C. neoformans* [123]. Hyaluronic acid is also embedded in the capsule and has been shown to be involved in penetration of the blood–brain barrier and subsequent cryptococcal meningitis [120].

Despite its importance for Cryptococcus virulence, the capsule is not the only virulence factor. Liu et al. [118] screened 1201 *C. neoformans* gene deletion strains in a mouse model of cryptococcosis and identified an additional 40 potential virulence factors that included replication factors, chromatin regulators, and immune response modulators. In addition, over 38 genes associated with melanin production were identified in a similar study. Melanin is thought to help protect the cells from oxidative stresses both in the host and in the environment [124]. The study identified novel genes involved in melanin production in vivo that were not previously linked to in vitro melanin production or Cryptococcus pathogenesis.

*C. neoformans* is also known to secrete a large number of enzymes with the potential to degrade host molecules [125]. The major enzymes involved in host toxicity are proteases, urease, phospholipase, and nuclease. *C. neoformans* can metabolize immunoglobulins and complement proteins, presumably through degradation by released proteases [126]. Thus, proteases may interfere with host defense mechanisms by cleaving immunologically important molecules or directly damaging effector cells. Furthermore, cryptococcal serine proteases promote increased BBB permeability [127], which may facilitate dissemination and subsequent brain infection.

Although gene deletion studies can test for the requirement of specific genes and their products in Cryptococcus pathogenesis, natural variation within the Cryptococcus population can also modulate gene sequences on a finer scale, affecting gene expression, RNA processing, or the function of the subsequent protein product. The role for fine-scale genetic differences is exemplified by studies that show differences in virulence between environmental and clinical isolates, as well as differences in both human and animal model virulence between clinical isolates within the same STs [128,129]. These results suggest that fine-scale variation can impact virulence, but studies that use limited sequencing to differentiate isolates, such as MLST, likely lack the resolution necessary to detect this fine-scale variation. Instead, whole-genome sequencing approaches will be critical to identify the role of fine-scale allelic differences, such as SNPs, that impact virulence.

## 7. *C. neoformans* Mixed Infection Genetic Diversity and Disease Outcome

To better understand the dynamics of initiation and progression in cryptococcal disease, it is important to study genetic and phenotypic differences in the context of human infection to identify the human and fungal risk factors that contribute to pathogenesis and poor clinical outcomes [19]. Of particular importance is the different clinical presentations and health outcomes that are associated with pathogenicity and virulence of Cryptococcus strains with respect to specific genotypes and phenotypes. Genetic diversity and analyses of susceptibility to antifungals in Cryptococcus isolates from patients are usually performed on single-colony isolates. The rationale behind this method is that the vast majority of patients are infected with a single genotype [130,131]. However, several studies have identified mixed infections in which individuals are infected with multiple genotypes [132,133]. However, our understanding of changes in Cryptococcus populations within patients throughout the course of treatment is still poorly understood.

A recent study analyzing five randomly selected single-colony isolates from 13 HIV patients showed that while the majority of patients harbored a single sequence type, 4 patients had mixed infections [133]. The patients with mixed infections experienced up to four shifts in isolate genotype at both the species and ST level across the course of the infection. This genetic diversity led to the co-existence of up to three Cryptococcus species and four different STs within the same individual during the infection [133]. An earlier study by Desnos et al. had similar results. In this study, analysis of between 4–33 single-colony isolates from each patient, as well as isolates from different organs, revealed mixed infections with different species or mating type were present in over 18% of the patients [132]. Another study also reported mixed infections within the same patient, with differences observed using microsatellite genotyping or AFLP fingerprinting [134]. All of these studies highlight that analysis of several isolates for each patient sample may be necessary to understand the diversity of genotypes and phenotypes that lead to disease presentation in patients. Assessment of genetic diversity is important, as studies in the human pathogenic fungal *Candida* spp. show that mixed infections can lead to treatment complications or failures, as well as the emergence of isolates resistant to antifungal drugs [135,136]. It is still unknown how much of the genetic diversity observed during mixed human Cryptococcus infections is due to co-infections with multiple different environmental strains or in vivo micro-evolution due to immune pressure or selection for antifungal drug resistance.

## 8. Associations between *C. neoformans* Genotypes and Clinical Presentation

As a preface to this section, it is important to clearly state that Cryptococcus phenotypes vary greatly across individual isolates. Differential attributes can be observed in association with genotype across broad phylogenetic relationship designations such as species, but also among more narrow designations such as molecular type or sequence type. For example, broadly comparing the species complexes, *C. neoformans* primarily causes infections in immunocompromised individuals, while *C. gattii* primarily causes infections in immunocompetent individuals [80]. However, under the umbrella of each species or sequence type are individual isolates with highly variable phenotypes. For example, some *C. deneoformans* clinical isolates exhibit higher virulence in mice than *C. neoformans* isolates even though in general *C. neoformans* isolates tend to cause more disease in both humans and in mouse models of cryptococcosis [19,46,133] Similarly, while the *C. gattii* species complex as a whole is generally associated with causing infection in immunocompetent individuals, some isolates predominantly cause disease in immunocompromised and HIV-infected individuals [80].

The mechanism underlying this large amount of individual isolate diversity in the human pathogenic *Cryptococcus* spp. remains unknown, but a few possibilities exist (Figure 1). First, each of the human pathogenic *Cryptococcus* spp. may have varying degrees of genetic diversity (for example, non-recombining vs. recombining sub-populations) which endow isolates with variable phenotypic or physiological attributes that could result in deviation from the broader species trends (Figure 1A). Alternatively, phenotypic differences between isolates could be due to micro-evolutionary events occurring in the human host or environmental niche that result in selection of isolate-specific genetic alterations that contribute to pathogenicity (Figure 1B). Finally, it is also possible that the high variability between isolates could be due to epigenetic differences between isolates that impact gene expression, virulence-related molecular functions and biological processes (Figure 1C). Importantly, even with this variability between individual isolates, MLST and GWAS analyses have begun to uncover evolutionary genetic relationships associated with human disease as well as traits associated with Cryptococcus survival within the environment.

Phylogenic studies using clinical isolates have found that various molecular types within the *C. neoformans* species, specifically VNB, exhibit vast genetic diversity [46]. In contrast, some of the species within the *C. gattii* species complex show minimal evidence of recombination [5,137,138]. With the advent of GWAS, evolutionary genetic changes that are associated with disease have been identified [7]. Similarly, a GWAS across cohorts of VNB isolates revealed sequence differences between clinical and environmental isolates in genes associated with virulence factors and stress responses [19]. Some of these genetic differences between molecular types may contribute to Cryptococcus phenotypes and clinical presentation.

Studies with *C. neoformans* African isolates have shown that isolates within the VNI molecular type are phenotypically associated with the production of “micro” cells that contribute to increased dissemination to the CNS [19,138]. Supporting this notion, patients infected with VNI isolates often present clinically with neurological symptoms, including vomiting and increased intracranial pressures [138]. In addition, both the VNI and the VNB subtypes are associated with differences in capsule shedding that can affect the host immune response [138]. For example, the VNB clade is associated with fever, whereas VNI is not. Interestingly, patients infected with VNB isolates have negative associations with neck stiffness and diastolic blood pressures but are positively associated with lumbar puncture opening pressures (increased intracranial pressures). Similar to VNI, VNB are both associated with lower CD4 counts, in agreement with the general trend of *C. neoformans* causing infection in immunocompromised patients [138]. In addition to cryptococcal meningitis, patients with VNB infections also tend to present with skin lesions and have higher mortality [5].

Sequence types (STs) have also been shown to influence infection through differences in phenotype, patient mortality and other clinical parameters of disease. Initial evidence of a relationship between host mortality and STs was observed among *C. neoformans* isolates collected during the Ugandan Cryptococcal Optimal ART Timing (COAT) trial [14]. The COAT isolates were categorized into three “virulence groups” based on human survival time and virulence in an animal model of cryptococcosis: (1) high virulence (ST93, ST40, ST31); (2) intermediate virulence (ST5, ST77, ST93); and (3) low virulence (ST5, ST40, ST31) [14]. Additional studies show that differential clustering of STs by severity of virulence phenotype is linked to an evolutionary divergence between the genetic sequences of the isolates [15]. For instance, clustering genetically similar clinical isolates into nonredundant evolutionary “burst groups” identified an association between ST93 mortality and immune response. In this analysis, Wiesner et al. [15] found that the ST93 Burst group 1 and Burst group 2 were associated with high patient mortality, while Burst group 3 had greater patient survival [15]. In addition, they found that Burst group 1 was also associated with increased capsule shedding, a known virulence factor in Cryptococcus [15].

Similarly, a study carried out in Zimbabwe demonstrated that *C. neoformans* genotypes demonstrated a high level of genetic diversity by microsatellite typing, and 51 genotypes within the molecular types VNI, VNB and VNII were identified. This study demonstrated that *C. neoformans* in Zimbabwe has a high level of genetic diversity when compared to global isolates [50]. Furthermore, genetic analysis showed that Cryptococcus strains found in Southern Africa represent a hotspot of genetic diversity. By combining this genetic data with microbiological analysis to assess virulence traits, the authors showed that genetic diversity is associated with differences in Cryptococcus phenotype. Finally, the authors analyzed detailed patient clinical data and showed that one genetic lineage (VNB) is significantly associated with survival [5].

To identify the specific single nucleotide polymorphisms (SNPs) or insertions/deletions (INDELs) that underlay this genetic diversity, Gerstein et al. performed the first genome-wide association study (GWAS) using Ugandan *C. neoformans* clinical isolates in 2019. This GWAS investigated genetic differences between ST93 isolates that were associated with patient outcome (Figure 2). The authors identified 145 non-synonymous SNPs and indels associated with 40 genes. Surprisingly, only two of these genes were previously known to be virulence determinants [7]. The fact that the majority of polymorphisms identified in this study were not in the canonical virulence factors highlights that the factors characterized to date allow Cryptococcus to be a pathogen, but that other undefined factors/genes differentiate isolates with high virulence from those with low virulence in humans. Finally, Clark et al. leveraged clinical, in vitro growth and genomic data for 284 *C. neoformans* isolates in a more recent GWAS to identify clinically relevant pathogen variants within patients with HIV-associated cryptococcosis in Malawi [46,137] (Figure 2).

These investigators also found both small and large-scale variations, including aneuploidy, which may impact the course of infection. Genes impacted by these variants were involved in transcriptional regulation, signal transduction, glycolysis, sugar transport, and glycosylation. The study went on to show that growth within the CNS was reliant upon glycolysis by using an animal model, and likely impacts patient mortality via CNS burden [18]. This study illustrates links between genetic variation and clinically relevant phenotypes, shedding light on survival mechanisms within the CNS and pathways involved in clinical persistence [7,18] Taken together, these findings demonstrate that there are associations between virulence traits (capsule phenotype and host mortality) and groups of STs that are genetically related [19].

## 9. Conclusions

Patient outcome depends on the interaction between the pathogen and the host. In sub-Saharan Africa, cryptococcal meningitis predominates as the cause of AIDS-related mortality. While it is known that the *C. neoformans* genotype impacts patient outcome, the main mechanisms underpinning this interaction are still not well understood. Understanding the mechanisms used by *C. neoformans* to facilitate virulence and adaptation to the host is necessary to better predict disease severity and establish proper treatment strategies. Importantly, we note that few of the studies elucidating these mechanisms, and using African isolates and patient data, were actually performed in African basic science laboratories. Thus, there remains a critical need to build the basic science infrastructure in Africa that will facilitate the rapid translation of these studies on the mechanism of Cryptococcus virulence into new strategies for clinical care.

## Figures and Tables

**Figure 1 jof-08-00734-f001:**
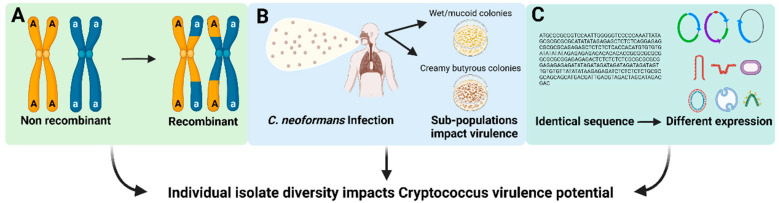
Factors that may contribute to individual isolate variation in Cryptococcus species. (**A**) Variation in pathogen genetic diversity. (**B**) Selection of isolate-specific genetic alterations with the host or environment. (**C**) Gene expression variation across isolates.

**Figure 2 jof-08-00734-f002:**
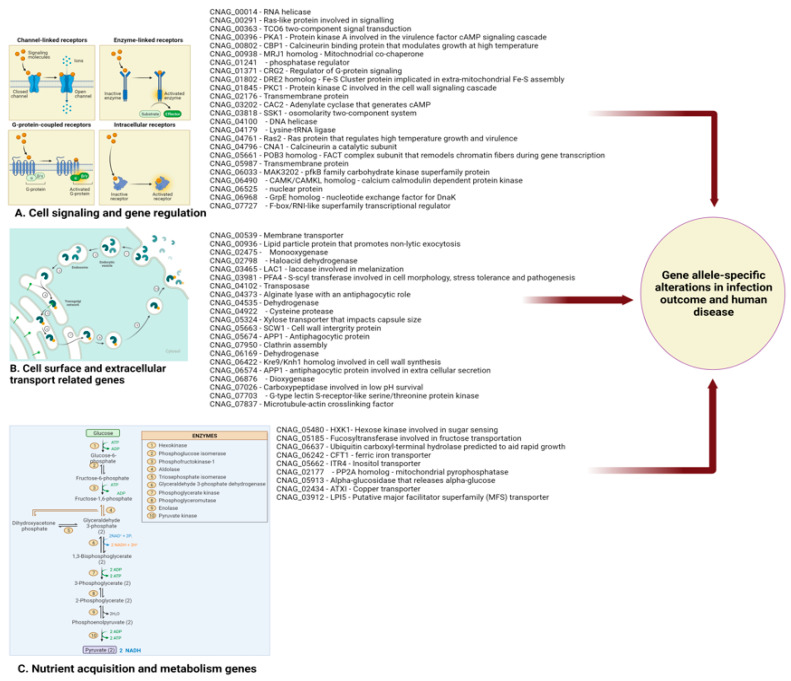
*C. neoformans* gene alleles identified in GWAS studies as impacting on human disease outcome. Created with BioRender.com.

**Table 1 jof-08-00734-t001:** Clinically relevant sequence types within the seven human pathogenic *Cryptococcus* spp. isolated from sub-Saharan African cryptococcosis patients.

Genus	Cryptococcus										
Species	*C. neoformans* Species Complex						*C. gattii* Species Complex				
	*C. neoformans*					*C. deneoformans*	*C. gattii*	*C. deuterogattii*	*C. bacillisporus*	*C. decagattii*	*C. tetragattii*
Lineages	VN1	VNII	VNBI	VNBII	VNIII	VNIV	VGI	VGII	VGIII	VGIIIVGIV	VGIV
Clinically relevant sequence types (ST)	**2** * 3 **4** * **6** * **9** * 23 **31** 36 39 58 **63** * 69 77 80 89 **93** * 20 22 90 311 317 379 380 449 450 483	40 42 43 207 208 233 262 334 467 555 576	9 18–21 43 210 245 249 261 263 384–387 392 394 396	409–411 415–417 419 421 424 428 429 432–434 436 438 447 451 460 464 465 472 478 504 537	N/A	11 112 121 160 260 557 578–580	51 58 106 162 208 215 490	5 7 8 12 20 25 31 44 46 75 172 173 321–324 445	59 64 75 79 84–86 89 93 142 164 209	N/A	69 70 221 491 492 493

* Bold indicates several sequence types that are observed frequently among clinical isolates in various geographical regions.

## Data Availability

Not applicable.

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
