# Peer review of "Cryptococcus neoformans Genotypic Diversity and Disease Outcome among HIV Patients in Africa"

_jof, 2022, doi:10.3390/jof8070734_

Round 1
Reviewer 1 Report
This is a nice review of factors contributing to C. neoformans infections and disease outcomes, including genetic heterogeneity of the strains and their correlation with phenotype.
1. I think adding a table to the manuscript summarizing the different genotypic factors that are associated with infection and/or disease would be a nice way to summarize the paper. This would include and summarize the last two paragraphs of section 7 (lines 389-419).
Minor points:
- On line 333, “analysis of 4-33 single colony isolates” is somewhat confusing. I suggest adding “between” before “4-33” to clarify the sentence.
- On line 375, I suggest changing “that” to “which” to avoid using “that” twice in the sentence and improving readability.
- The title in reference 19 needs to be fixed.
- References 80 and 83 are the same paper.
- I think references 99 and 100 are also the same paper.
Reviewer 2 Report
I congratulate the authors for this interesting review on aspects of the pathogenicity of yeasts of the genus Cryptococcus and the disease they produce, cryptococcosis. However, the title of the paper suggests that it is a review of special aspects described in isolates of African origin, which is a very interesting topic. However, the paper hardly analyses or comments on peculiarities of these isolates. Thus, the most important comment on the paper is that it is not a true review of African isolates, and it should be. Therefore, my suggestion to the authors is that they try to redirect the review of the subject, focusing the comments on teh african isolates described in exisiting works (phenotypic and genotypic variety; virulence, clinical outcome, response to treatment, etc)
Other details to be revised are discussed below:
- The authors mention on at least two occasions, that infection is acquired by inhalation of spores from the environment (see lines 103 and 117). Although this is correct, as far as I have information, we do not know whether the infection is caused exclusively by inhalation of spores (basidiospores), by inhalation of dried yeast cells, or both. Therefore, this comment should be added (infectious spores or desiccated cells from the environment / inhalation of aerosolized basidiospores and/or desiccated yeasts cells).
- The authors state that the most common cryptococcal infection is asymptomatic (see lines 115-116) and that there is evidence that human contact with Cryptococcus is a frequent phenomenon, as demonstrated by Goldman in 2001 and others (lines 107-108). In my opinion it should be mentioned here that Cryptococcus is one of the yeasts most frequently described as members of the pulmonary mycobiome. I recommend reading the article by Rubio-Portillo, The domestic environment and the lung mycobiome (Microorganisms 2020, 8(11), 1717. DOI: https://doi.org/10.3390/microorganisms8111717), which agrees of this and demonstrates this phenomenon. In addition, it shows that people who have Cryptococcus DNA in the lower airway, also tend to have it detectable in their domestic environment.
- The article hardly relies on iconography, which is often helpful in facilitating the understanding of its content. I encourage the authors to include some figures showing relevant aspects of African strains. For example, the most frequently described STDs, with respect to their distribution in other areas of the world or, within those represented in Figure 2, which are the most prevalent in Africa. Additionally, for figure 3, I consider that the representation of the expression of different phenotypes is not very accurate, as the drawing seems to indicate different sequences of the genotype.
- Finally, in the references, number 10 and 103 are the same, they are repeated and one of them should be deleted and the citations in the text have to be corrected accordingly.
